# The Oral Microbiome across Oral Sites in Cats with Chronic Gingivostomatitis, Periodontal Disease, and Tooth Resorption Compared with Healthy Cats

**DOI:** 10.3390/ani13223544

**Published:** 2023-11-16

**Authors:** Jamie G. Anderson, Connie A. Rojas, Elisa Scarsella, Zhandra Entrolezo, Guillaume Jospin, Sharon L. Hoffman, Judy Force, Roxane H. MacLellan, Mike Peak, Bonnie H. Shope, Anson J. Tsugawa, Holly H. Ganz

**Affiliations:** 1Department of Oral Medicine, Penn Dental Medicine, Philadelphia, PA 19104, USA; 2AnimalBiome, Oakland, CA 94609, USA; connie@animalbiome.com (C.A.R.); elisa@animalbiome.com (E.S.); zhandra@animalbiome.com (Z.E.); guillaume@animalbiome.com (G.J.); holly@animalbiome.com (H.H.G.); 3Veterinary Dental Consulting, Jacksonville, FL 32226, USA; shoffmandvm@gmail.com; 4Dentistry for Animals, Aptos, CA 95003, USA; 5VCA Highlands Ranch Animal Specialty and Emergency Center, Highlands Ranch, CO 80126, USA; rmaclell@yahoo.com; 6The Pet Dentist, Inc., Tampa, FL 33544, USA; thepetdentist@tampabay.rr.com; 7Veterinary Dental Services LLC, Boxborough, MA 01719, USA; bshope@veterinarydental.com; 8Dog and Cat Dentist, Inc., Culver City, CA 90232, USA

**Keywords:** gingivostomatitis, stomatitis, oral microbiome, dysbiosis, periodontal disease, tooth resorption, cats

## Abstract

**Simple Summary:**

Feline chronic gingivostomatitis (FCGS) remains a poorly understood clinical condition with significant impact on the quality of life of affected cats. An understanding of the pathogenesis of this inflammatory disease will enable the development of improved and targeted therapeutics beyond full-mouth extractions. Here, we collected sterile noninvasive plaque samples at ten distinct sites within the oral cavity in cats with FCGS (*n* = 12), healthy cats (*n* = 9), or cats with other oral conditions (*n* = 11). We used 16S rRNA gene sequencing (V1–V9) to profile bacteria in the oral microbiome. We found that the microbiomes of cats with FCGS were distinct from those of healthy cats at multiple oral sites, indicating that dysbiosis is present throughout the oral cavity. The microbiomes of cats varied depending on their oral diagnoses, confirming that the various dental diseases impact the microbiome in different ways. Lastly, microbiome data obtained from swabs of the oral cavity were similar to those obtained using endodontic paper point plaque samples, suggesting this approach as another valuable method of sampling. Given these additional insights, future studies can focus on targeted therapeutics so that extraction of healthy teeth will no longer be the standard of care for this challenging condition.

**Abstract:**

Feline chronic gingivostomatitis (FCGS) is a chronic mucosal and gingival inflammatory disease in which pathogenesis remains unclear. Interactions between the host inflammatory process, the host immune response, and the oral microbiome are implicated in this pathogenesis. To begin to understand this disease and the impact of the microbiome to host inflammatory disease states, we collected sterile noninvasive plaque biofilm samples from ten distinct sites within the oral cavity in cats with stomatitis (*n* = 12), healthy cats (*n* = 9), and cats with tooth resorption or periodontitis (*n* = 11). Analysis of full-length 16S rRNA gene sequences indicated that the microbiomes of cats with FCGS presented marked dysbiosis at multiple oral sites. Additionally, microbiome beta diversity varied with oral condition, indicating that stomatitis, periodontitis, and/or tooth resorption influence the microbiome differently. Lastly, we found that the microbiomes of swabs taken from the oral cavity were comparable to those taken from plaque using endodontic paper points, validating this as another sampling method. Collectively, our work furthers our understanding of the dysbiosis and composition of bacteria in the oral microbiome in FCGS, with hopes of contributing to the prevention, diagnosis, and treatment of this challenging condition in felines.

## 1. Introduction

Feline chronic gingivostomatitis (FCGS) is a chronic immunoinflammatory disease process with microbiome influences [1,2,3]. The exact pathogenic mechanisms are unclear, and crosstalk between the microbiome, host immune cells, and interactions with other concomitant inflammatory disorders like periodontal disease (PD) and external tooth resorption (TR) are beginning to be understood [1,2]. Similar to inflammatory bowel disease, the pathogenesis of FCGS may vary due to the diversity of its presentations and cofactors. Presentations include ulcerative lesions versus proliferative tissues, presence or absence of caudal stomatitis, presence or absence of other oral inflammatory disorders, and those rarer cases with no history of dental disease [4,5]. The clinical presentation of FCGS generally dictates the diagnosis and most often the lesions are bilaterally symmetrical. Lesion locations are variable and can involve the gingiva, mucosa, tongue, and caudal oral mucosa. The presentations can have variable therapeutic responses as well. Cofactors in FCGS include feline calicivirus (FCV) [6], feline immunodeficiency virus (FIV) [7], and *Bartonella* infection [8]. The presence of other inflammatory disorders could also be important; a large FCGS case series found that cases with TR had a more favorable outcome, and that both PD and TR were associated with alveolar bone loss. From a molecular standpoint, transcriptome (e.g., gene expression of Janus kinase (JAK)-signal transducer and activator of transcription (STAT) pathway, interleukin-17, interleukin-6) and proteome (e.g., proinflammatory cytokines) biomarkers of FCGS were recently reported [9,10].

Next-generation sequencing has revolutionized our understanding of the microbiome in veterinary oral medicine. Several investigations have described the microbiomes of cats with FCGS or periodontitis. In most of these studies, the caudal mucosa or gingiva were chosen as the most representative sites to sample. These studies have revealed that healthy subgingival samples are dominated by the bacterial genera *Porphyromonas*, *Enhydrobacter*, *Capnocytophaga*, *Moraxella*, and *Fusobacterium* [11]. Healthy cats can also harbor greater abundances of *Actinomyces*, *Bergeyella*, *Dichelobacter*, *Comamonas*, *Actinobacillus*, and *Myroides* than cats with chronic periodontitis, aggressive periodontitis, or FCGS [11].

Conversely, cats diagnosed with FCGS have oral microbiomes where *Porphyromonas gulae*, *Peptostreptococcus canis*, *Treponema* spp., *Fretibacterium* spp., and *Propionibacteriaceae* spp. are abundant [11,12]. Their oral microbiomes may also be enriched in *Fusobacterium nucleatum* [12] or *Odoribacter* compared with healthy cats [13]. *T. denticola* in particular has been associated with irreversible periodontal disease in companion animals [14,15]. Additionally, in dogs with severe periodontal disease, *Methanobrevibacter oralis*, *P. canis*, and *Tannerella* spp. are part of the core microbiome found across individuals [16]. In canine chronic ulcerative stomatitis, the bacterial microbiome of the ulcerative lesion is dominated by *Porphyromonas* spp., *Neisseria weaveri*, *Fusobacterium* spp., and a *Tannerella forsythia*-like phylotype [17].

Nonetheless, it is unknown whether dysbiosis in the oral microbiomes of cats with FCGS extends to multiple oral biogeographical sites, and whether differences among biogeographical sites are observed. Evaluation of other oral sampling sites, such as the tongue dorsum, sublingual surface, buccal mucosa, and salivary molar gland, have not previously been investigated, and may highlight controversial and diverging hypotheses. Here, we address these gaps in knowledge and use full-length 16S rRNA gene sequencing to characterize the microbiomes of cats with FCGS (*n* = 12) and clinically healthy cats (*n* = 9) at ten distinct sites in the oral cavity. We ask whether the microbiomes partition by body site within the two groups and whether they differ between groups. Furthermore, because it is unknown whether the disease mechanisms or microbiomes of cats with different oral diseases are the same across sites, we also study the microbiomes of cats with severe PD (*n* = 5) and with idiopathic TR (*n* = 6), and compare them with the microbiomes of cats diagnosed with FCGS. Lastly, because not many studies have discussed sampling methods that are appropriate for collecting oral plaque, we compare the microbiomes obtained from plaque using endodontic paper points with those obtained using swabs of the oral cavity. Our work furthers an understanding of the dysbiosis and composition of the oral microbiome in FCGS, with hopes of contributing to future translational investigations, biomarker discovery, and targeted therapies.

## 2. Methods

### 2.1. Ethics Statement

Standard veterinary private practice hospitals, as opposed to Veterinary Medical Teaching Hospitals, do not employ Institutional Animal Care and Use Committee (IACUC) protocols. As such, we utilized and conformed our study procedures in accordance with the American Animal Hospital Association Guidelines for Dental Care [18]. Additionally, the Academy for Veterinary Dentistry (https://www.avdonline.org/, accessed on 30 October 2021) approved our grant proposal (and the ethics of such) in awarding funding in March 2021. Client consent forms detailing the study were discussed and signed. No adverse events were documented as a result of plaque sampling.

### 2.2. Clinical Evaluations of Feline Participants

The microbiomes of the oral cavity at ten distinct biogeographical sites were sampled noninvasively from thirty-two (*n* = 32) privately owned domestic cats presenting to six specialty veterinary hospitals for a routine comprehensive oral health assessment and treatment (COHAT) [19], or for diagnosis and treatment of FCGS. The cats were negative for feline leukemia virus (FeLV) and feline immunodeficiency virus (FIV), did not have any systemic disease issues (diabetes mellitus, inflammatory bowel disease), and had not received antibiotics or immune modulating drugs within the month prior to sampling. The cats were deemed overall clinically orally healthy with no signs of any oral disease (*n* = 9), or were diagnosed with FCGS (*n* = 12), severe periodontal disease (*n* = 5), and/or external tooth resorption (*n* = 6). Of the twelve FCGS cats, six had TR and four had SPD. Periodontal status, evidence of tooth resorption, and FCGS diagnosis were assessed by a Board Certified Veterinary Dentist™ based on clinical examination, periodontal examination, and full-mouth dental radiographs [20,21].

Information on these clinically validated samples, including the feline patients’ age, sex, weight, and breed, is in Appendix A and summarized in Table 1.

### 2.3. Microbiome Sampling

Sampling of plaque biofilm was accomplished by sweeping endodontic paper points along a plaque retentive surface while the cat was under general anesthesia. Anesthetic protocols varied between veterinary dental specialists but broadly involved administering the induction drugs alfaxalone 2–4 mg/kg or propofol 2–4 mg/kg, both with diazepam 0.25 mg/kg. Sampling was performed prior to the use of any oral irrigants. The ten sites sampled were three mucosal sites (caudal, buccal, and normal), two tongue surfaces (dorsal and sublingual), two sites associated with disease (external tooth resorption site for TR cats and deep periodontal pocket for SPD cats), and the gingival sulcus, molar flap, and tooth surface. Swabs were also taken from the oral mucosa of study participants to obtain a general view of the microbiome in the mouth. Upon collection, both paper point and swab samples were placed into 1.5 mL microcentrifuge tubes and kept at −80 °C until shipment overnight on dry ice to AnimalBiome (Oakland, CA, USA).

### 2.4. DNA Extraction and Full-Length 16S rRNA Amplicon Gene Sequencing

Genomic DNA was extracted from paper point samples using the QIAGEN DNeasy Powersoil Pro Isolation Kit on the QIAcube high-throughput (HT) instrument (QIAGEN, Redwood City, CA, USA), following the manufacturer instructions. DNA concentration was measured using Qubit dsDNA High-Sensitivity Kit (ThermoFisher, Waltham, MA, USA). Primers 27F (5′-AGRGTTYGATYMTGGCTCAG-3′) and 1492R (5′-RGYTACCTTGTTACGACTT-3′), tailed with 16 bp asymmetric barcode sequences, were used for full-length (V1 to V9) 16S rRNA gene amplification [1]. PCR amplification was performed using 12.5 μL of KAPA HiFi HotStart ReadyMix PCR kit (KAPA Biosystems, Wilmington, MA, USA), 3 μL of both Forward and Reverse primers (2.5 μM), 5 μL of template DNA, and PCR-grade water required for a final volume of 25 μL. PCR conditions used were initial denaturation at 95 °C for 3 min and 27 cycles of denaturation at 95 °C for 30 s, annealing at 57 °C for 30 s, and extension at 72 °C for 60 s. The purified amplicons were sequenced using PacBio Sequel IIe chemistry (Pacific Biosciences, Menlo Park, CA, USA).

### 2.5. Bioinformatic Processing of PacBio CSS Reads

After sequencing, circular consensus sequencing reads (CCS) were converted to HiFi reads for each demultiplexed sample using Single Molecule, Real-Time (SMRT) Link Application software (v.11.0.0.146107). The resulting reads were trimmed, denoised, dereplicated, and filtered of chimeras using Quantitative Insights Into Microbial Ecology (QIIME) 2 [22] and the Divisive Amplicon Denoising Algorithm (dada2) plugin [23] as outlined in AnimalBiome’s tutorial for processing and classifying PacBio full-length 16S rRNA HiFi reads (https://github.com/AnimalBiome/AB_FlexTax/tree/main, accessed on 3 August 2023). Briefly, the pseudo-pooling method was used for denoising, the maximum number of expected errors was set to 3, and reads shorter than 1300 bp or longer than 1600 bp after adapter trimming were removed. Samples were pooled for chimera detection.

In preparation for taxonomic annotation of amplicon sequence variants (ASVs), we manually curated the Silva (v.138.1 NR99) reference database [24] to remove sequences that did not have mostly full-length versions of the 16S rRNA gene (e.g., were <1300 bp or >1600 bp), were nonprokaryotic (e.g., had a species label of *Chrysanthemum morifolium*), or had an uninformative species label (e.g., were “uncultured bacterium”, “proteobacterium”, or “freshwater sediment”). Refer to the tutorial mentioned above for the full list of excluded sequences. These steps reduced the size of the Silva database from 510,508 sequences to 116,173 sequences, and from 80,381 unique taxa to 17,746 unique taxa.

As outlined in the tutorial, taxonomic assignment of ASVs was primarily performed using a Naive Bayes trained sklearn classifier in QIIME2, setting a confidence threshold of 0.7. These labels were further refined using stringent VSEARCH [25] classification, also in QIIME2. This dual hybrid approach allowed us to obtain greater specificity in our taxonomic labels (e.g., if an ASV was assigned genus-level classification by sklearn but VSEARCH assigned it to species with 100% confidence, then we retained the species-level call).

### 2.6. Statistical Analyses of Microbiome data: Healthy Cats Compared with Cats Diagnosed with FCGS

All statistical analyses were performed using the R statistical software program (v4.3.0) [26]. For this, we imported the sample metadata (Appendix A), ASV counts table (Appendix A), and list of ASV taxonomic classifications (Appendix A) into R. Any samples with fewer than 100 reads after sequence processing were not included in our analyses; this resulted in the exclusion of sixty-nine samples.

Because of this, not all cats had a complete set of 10 plaque samples each. In fact, seven cats had 9 plaque samples, two cats had 8, seven cats had 7 samples, another seven cats had 6, four cats had 5 plaque samples, four cats had 4, and one cat only had 3 sites represented (Appendix A). The samples that were included in the analyses had an average sequencing depth of 15,210 reads and a median sequencing depth of 8701 reads (range: 106 to 161,133 sequences). Similarly, not all cats had accompanying swab samples from the oral cavity. This was true of three cats diagnosed with TR and one healthy cat. Swab samples had an average sequencing of 10,954 sequences and median sequencing depth of 10,891 (range: 5290–18,314).

To visualize microbiome community composition across biogeographical sites in healthy cats and cats with FCGS, we generated stacked bar plots showing the relative abundances of bacterial genera or species across samples using ggplot2 (v3.4.2) [27]. Bacterial taxa with mean relative abundances < 1.25% were clumped into an “Other” category.

To estimate microbiome alpha diversity (within-sample diversity), we first rarefied samples to 7000 sequences using the GUniFrac R package (v1.7) [28] to account for differences in library depth. This resulted in the exclusion of 102 samples and the retention of 136 samples. Two metrics of alpha diversity were computed based on ASV counts: ASV richness (observed richness) (phyloseq package v1.44.0) [29] and Pielou’s evenness (microbiome package v1.22.0) [30]. Generalized linear models (GLMs) using a Poisson distribution (for ASV richness) or quasibinomial distribution (for Pielou’s evenness) tested whether microbiome alpha diversity was different between healthy cats and cats diagnosed with FCGS. The model term contained information on both the oral site and the group (e.g., “FCGS-Caudal Mucosa” or “Healthy-Sublingual”) and pairwise comparisons with the emmeans package were restricted to within an oral site (e.g., “FCGS-Caudal mucosa” vs. “Healthy-Caudal mucosa”). *p*-values were adjusted for multiple comparisons using the false discovery rate (FDR) method. Boxplots of microbiome alpha diversity were made with ggplot2.

For beta diversity analyses (estimates of dissimilarity between samples), the ASV table was converted to proportions for Bray–Curtis distances, or applied a center log ratio (CLR) transformation for Aitchison distances. To test whether the microbiomes of healthy cats and the microbiomes of cats with FCGS were distinct, we used permutational multivariate analysis of variance (PERMANOVA) models with 999 permutations (vegan package v2.6-4 [31]). Each oral site was tested independently, and all *p*-values were adjusted with FDR. PCoA ordinations faceted by oral site and color-coded by sample type (healthy vs. FCGS) were made with ggplot2.

PERMANOVAs also tested whether oral microbiomes varied by site location in the mouth. The global model included oral site and host identity (to account for the repeated sampling of individuals) as predictors. One model was employed for the healthy samples, and a separate model was employed for the FCGS samples. Because the global models revealed statistically significant *p*-values, pairwise comparisons using the pairwise Adonis package (v0.4.1) [32] revealed which oral sites were different from each other. *p*-values were adjusted for multiple comparisons using the FDR method.

Differential abundance analyses with the R LinDA package (v0.1.0) [33] were employed to highlight any bacterial species that were enriched in healthy cats compared with cats diagnosed with FCGS (or vice versa). For this, samples from all oral sites except the gingival sulcus and dorsal tongue were pooled, and models included a random effect of host identity. The prevalence cutoff was set to 20%, winsorization cutoff (quantile) to 0.97, and *p*-value adjustment to “FDR”. Identical LinDA analyses were repeated for bacterial genus abundances.

### 2.7. Statistical Analyses of Microbiome Data: Cats Diagnosed with FCGS, TR, or SPD

The second part of this study involved examining whether the microbiomes of cats diagnosed with the different oral conditions (FCGS, FCGS_SPD, FCGS_TR, TR, SPD) differed from each other. Generalized linear mixed models (GLMMs) with the lme4 package (v1.1-33) [34] tested whether microbiome alpha diversity varied with oral condition. Samples from all oral sites were combined for this analysis due to uneven sample sizes at each site. The models set host identity as the random effect. The statistical significance of the main effects in the models were quantified via likelihood ratio tests (LRTs). Post hoc comparisons were performed with the emmeans (v1.8.7) [35] and multcomp (1.4–23) packages [36], and *p*-values were adjusted with Tukey’s method.

An identical model was outlined for beta diversity analyses. PERMANOVAs tested whether microbiome differences were observed among the cats with the varying oral conditions. Pairwise comparisons with the pairwise Adonis package illustrated that all conditions were significantly different from each other.

One last PERMANOVA model converted the oral condition variable into three Boolean variables: has FCGS (Y/N), has SPD (Y/N), or has TR (Y/N). This would mean that a cat with FCGS and periodontitis would be a “Yes” for both the FCGS and SPD Boolean variables. This was done to determine which of the three conditions was the most influential on the microbiome. The six oral sites with the largest sample sizes were examined.

### 2.8. Statistical Analyses of Microbiome Data: Swab Samples vs. Plaque Samples

The third objective of this study was to compare the microbiome data obtained using swabs from the oral cavity and plaque samples from specific sites in the oral cavity. We chose the gingival sulcus as our plaque site as this site had an almost complete set of samples.

Stacked taxa bar plots visualized the relative abundances of bacterial genera across swab samples and plaque samples. PERMANOVA tests determined whether there was a correlation between the two sample types. They set host identity (i.e., pet name) and sample type (i.e., swab vs. paper point) as predictors. Linear mixed models (LMMs) were used for alpha diversity analyses and included the same two predictors.

A second PERMANOVA model correlated microbiome beta diversity with host age (years), sex (M/F), body condition (kg), and oral condition (FCGS, FCGS_SPD, FCGS_TR, SPD, TR).

## 3. Results

### 3.1. Feline Participants

Of the thirty-two cats that participated in this study, 46% were male and 54% were female, excluding five cats that had missing metadata values (Table 1 and Appendix A). All cats were spayed or neutered. They ranged in age from 1.5 to 14.6 years old (mean: 6.7 years, median: 6.4 years) and weighed between 2.8 kg and 7.4 kg (mean: 5 kg, median: 4.75 kg) (Table 1). Ages and body weights of cats diagnosed with FCGS were not distinct from those of healthy cats or cats diagnosed with only TR or severe periodontitis (age GLM LRT χ^2^ = 0.96, *p* = 0.61; body weight GLM LRT χ^2^ = 0.91, *p* = 0.63). However, the median age was higher for cats with TR compared with cats in the other categories (Table 1).

### 3.2. Is Microbiome Dysbiosis Observed in Cats with FCGS?

The purpose of this study was to characterize the biogeography of the oral microbiome in cats with chronic gingivostomatitis (FCGS), healthy cats, and cats with two other inflammatory diseases (severe periodontal disease—SPD and tooth resorption—TR). We investigated whether the microbiomes of cats afflicted with FCGS showcased bacterial dysbiosis and were distinct from those of healthy animals at multiple oral sites.

We found that overall the microbiomes of cats with FCGS presented marked dysbiosis at multiple oral sites (Figure 1A–D) and had elevated abundances of *Porphyromonas* (particularly *P. gulae* 13.42% mean relative abundance, *P. macacae* 4.22%, and *P. circumdentaria* 4.13%), *Bacteroides* (mainly *B. pyogenes* 3.56%), *Treponema* (most abundant was *T. denticola* 1.12%), *Tannerella* (all *T. forsythia* 1.36%), *Peptostreptococcus* (mainly *P. canis* 3.4%), and *Fusobacterium* (most abundant was *F. russii* 2.33%) compared with the oral microbiomes of clinically healthy cats (Figure 1A–D, Appendix A). They also had reduced abundances of unclassified *Moraxella* (0.75% FCGS vs. 6.02% healthy), *Conchiformibius* (mainly *C. kuhniae*; 2.11% FCGS vs. 5.17% healthy), and *Frederiksenia* (0.36% FCGS vs. 2.86% healthy) (Figure 1A–D, Appendix A).

Alpha diversity analyses indicated that the microbiomes of cats with FCGS were less rich in bacterial taxa than the microbiomes of healthy cats at all oral sites except for the normal mucosa and the tooth surface (LRT GLM observed richness χ^2^ = 2215, *p* < 0.0001; see Appendix A for post hoc comparisons) (Figure 2A). The two groups of cats did not differ in their microbiome evenness (LRT GLM Pielou’s evenness χ^2^ = 20.35, *p* = 0.15; see Appendix A for post hoc comparisons) (Figure 2B).

Microbiome dysbiosis in FCGS cats was confirmed via beta diversity analyses as well. The oral microbiomes of healthy cats and cats diagnosed with FCGS were significantly different from each other at all oral sites except for the normal mucosa (pairwise PERMANOVAs *p* < 0.05; see Appendix A) (Figure 2C, Appendix A). This suggests that dysbiosis in these oral bacterial communities is not only evident at gingival and mucosal sites, but also in less studied sites like the tongue, molar flap, and tooth surface.

To identify the specific bacteria that may be driving this dysbiosis, we conducted differential abundance testing. For this, we first determined whether we could pool microbiome data across oral sites; this pooling would be conducted if the oral sites did not differ from each other. Within the clinically healthy samples, oral microbiomes did partition by oral site, and this factor accounted for up to 23% of the variation (two-term PERMANOVA Bray–Curtis R^2^ = 0.23, *p* = 0.001; Aitchison distance R^2^ = 0.19, *p* = 0.001). Post hoc tests indicated that differences were mainly concentrated between the gingival sulcus and the other body sites, and between the dorsal tongue and the other sites (Figure 3, Appendix A). Additionally, there was consistency in the composition of oral microbiomes from the same individual; samples from the same cat were similar to each other despite coming from different areas of the mouth (PERMANOVA Bray–Curtis R^2^ = 0.28, *p* = 0.001; Aitchison distance R^2^ = 0.33, *p* = 0.001).

The microbiomes of cats diagnosed with FCGS also clustered by oral site, but this factor accounted for much less of the variation (10%) (two-term PERMANOVA Bray–Curtis R^2^ = 0.098, *p* = 0.001; Aitchison distance R^2^ = 0.076, *p* = 0.001). Post hoc tests revealed that differences mainly lay between the dorsal tongue and the other sites (Figure 3, Appendix A). Host individual identity was an even stronger predictor of microbiome beta diversity (PERMANOVA R^2^ = 0.54, *p* = 0.001; Aitchison distance R^2^ = 0.51, *p* = 0.001). These analyses indicated that it was statistically appropriate to pool data from all sites (except the gingival sulcus and the dorsal tongue) for differential abundance analysis.

Differential abundance analysis indicated that twenty-one bacterial species were underrepresented in the oral microbiomes of cats with FCGS compared with those of healthy cats. Among these bacteria were *Bergeyella zoohelcum*, *Canibacter oris*, *Capnocytophaga canimorsus*, *Conchiformibius kuhniae*, *Corynebacterium mustelae*, *Frederiksenia canicola*, two *Neisseria* species, *Pauljensenia hongkongensis*, and *Streptobacillus felis* (Figure 4A, Appendix A). The microbiomes of cats with FCGS were only enriched in *Flexilinea flocculi* (Figure 4A, Appendix A). When examining the abundances of bacterial genera, the results showed that the microbiomes of healthy cats harbored larger abundances of 27 bacterial genera, including *Flavobacterium*, *Moraxella*, *Conchiformibius, Neisseria*, *Bergeyella*, *Streptococcus*, *Catonella*, *Actinobacillus*, and *Cardiobacterium* (Figure 4B, Appendix A). The results suggest that at least in our dataset, dysbiosis is characterized by the loss or reduced abundances of beneficial or commensal bacteria, rather than the enrichment of a particular pathogen.

### 3.3. Do the Microbiomes of Cats with Different Oral Conditions Diverge from Each Other?

After comparing the microbiomes of cats with FCGS and healthy cats, we also wanted to see if the microbiomes of cats with different oral conditions diverged from each other. Oral microbiome evenness but not richness varied with oral condition (LRT GLMM ASV richness χ^2^ = 3.61, *p* = 0.46; LRT Pielou’s evenness χ^2^ = 21.07, *p* < 0.01) (Figure 5B). The oral microbial communities of cats diagnosed with both FCGS and severe periodontal disease (FCGS-SPD) were less diverse than the oral microbiomes of cats with the other oral conditions (Figure 5B, Appendix A).

Similarly, for beta diversity analyses, differences between all groups were apparent (PERMANOVA Bray–Curtis R^2^ = 0.134, *p* = 0.001; Aitchison distance R^2^ = 0.108, *p* = 0.001; all pairwise comparisons had adjusted *p*-values < 0.05) (Figure 5C). Oral diagnosis accounted for up to 13% of the variation in the microbiome.

Next, we investigated which of the three oral conditions (FCGS, TR, or SPD) most strongly predicted microbiome beta diversity. For this, we coded the three conditions as binary variables (e.g., yes SPD vs. no SPD); this analysis would highlight which oral sites are more heavily impacted by a particular oral condition. An FCGS diagnosis was more influential in predicting microbiome beta diversity in the molar flap, dorsal tongue, and tooth surface (Appendix A). Periodontitis was more influential in predicting microbiome beta diversity in the gingival sulcus (Appendix A). Interestingly, caudal and normal mucosal microbiomes were not strongly correlated with the presence or absence of stomatitis, periodontitis, or tooth resorption (Appendix A).

Unfortunately, differential abundance analyses could not point to any particular bacterial taxon as driving differences between cats with FCGS and cats without FCGS, or cats with SPD and cats without SPD (all bacterial taxa had adjusted *p*-values > 0.05; results not shown).

### 3.4. Do the Microbiomes of Swabs from the Oral Cavity Approximate Plaque Microbiomes Obtained Using Endodontic Paper Points?

Twenty-seven cats had microbiome data from both plaque samples and swab samples. We determined whether swabs from the oral cavity approximated plaque samples taken from the gingival sulcus.

Analyses of microbiome beta diversity revealed that overall there was a high degree of fidelity between the two sample types; that is, samples from the same cat were similar to each other despite coming from different sources (Figure 6B,C). Host identity accounted for up to 73% of the variation in the microbiome, while sample type only accounted for 3% of the variation (PERMANOVA Bray–Curtis sample type R^2^ = 0.031, *p* = 0.002; host identity R^2^ = 0.709, *p* = 0.001; PERMANOVA Aitchison sample type R^2^ = 0.024, *p* = 0.002; host identity R^2^ = 0.738, *p* = 0.001). Plots of microbiome composition (Figure 6A) showed that there were compositional similarities between the two sample types, with some differences in the relative abundances of certain bacterial groups which was expected.

Analyses of microbiome alpha diversity indicated that the microbiomes from swab samples were slightly more even than but equally rich as the microbiomes from paper point samples (LMM χ^2^ = 0.24, *p* = 0.61; LRT GLMM χ^2^ = 4.9, *p* = 0.026). The mean microbiome evenness for swab samples was 1.94, while plaque samples had a mean evenness of 1.74.

Lastly, we found that the microbiomes from swab samples but not plaque samples were significantly correlated with host sex. Neither sample type was correlated with the cat’s age or weight (kg) (Table 2). Swab samples also varied with oral condition (e.g., FCGS_SPD, SPD, TR) (PERMANOVA Bray–Curtis R^2^ = 0.25, *p* = 0.04; Aitchison R^2^ = 0.27, *p* = 0.006). Collectively, these findings suggest that swabs of the oral cavity capture similar amounts of variation in the oral microbiomes of cats as do plaque samples taken with endodontic paper points.

## 4. Discussion

Since its first description in 1977 [37], feline chronic gingivostomatitis has been a debilitating enigma for feline patients. Various etiologic agents have been implicated in initiating and maintaining the disease. And controversy still exists as to its exact pathogenesis, whether inflammatory, viral, microbiome-mediated, or metabolic/immune-dysregulated. As a result, prognosis and therapy remain difficult and nonscientific.

With advances in sequencing, the research community has made strides in elucidating the bacteria present in the oral cavities of cats with FCGS and thus have begun to speculate on their role in the pathogenesis of this disease. Much of the sentinel work on FCGS has focused on a particular site in the oral cavity: the caudal oral mucosa (also known as the palatoglossal arch or fauces). Historically this site has been thought to be more representative of the disease and associated with a worse prognosis, while sites such as the tongue are thought to be least involved in FCGS. In truth, it was not known whether the palatoglossal arch accurately represents the disease entity or whether other oral sites can also be sampled to represent the oral microbiome.

To address these gaps in knowledge, we surveyed the oral microbiomes of cats with chronic gingivostomatitis at ten sites within the oral cavity and compared these with the microbiomes of healthy cats. Furthermore, because many previously published studies did not examine different forms of FCGS, we compared the oral microbiomes of cats with FCGS, SPD, TR, or a combination of these conditions and determined whether differences in the microbiome exist between the conditions. Lastly, not many studies have discussed the sampling methods that are appropriate for collecting oral biofilm or plaque. We analyzed the microbiomes obtained from plaque using endodontic paper points and those obtained using swabs of the oral cavity. We confirmed that they are similar, and that microbiome variation and diversity can be sufficiently captured by swab samples.

### 4.1. Bacteria That Are Abundant in the Oral Cavities of Cats with FCGS

The results showed that across oral sites, the microbiomes of cats with FCGS were dominated by *Porphyromonas gulae* (13% of microbiome), *Porphyromonas macacae* (4.2%), *Porphyromonas circumdentaria* (4.13%), *Bacteroides pyogenes* (3.5%), *Peptostreptococcus canis* (3.4%), and *Fusobacterium russii* (2.3%), among others. Similarly, a prior study reported that the most abundant taxa in the oral microbiomes of cats with FCGS were *P. gulae* (16.9%), *P. canis* (4.9%), *Bacteroidales* spp. (3.8%), *Fretibacterium* spp. (3.5%), and *Propionibacteriaceae* spp. (2.7%) [12]. The findings are not completely overlapping given that the prior study utilized swab samples and 16S rRNA gene sequencing of the V1-V3 region whereas we sampled plaque at specific locations, and used full-length 16S rRNA gene sequences. Another study reported that *P. gulae* and *P. circumdentaria* were the two most frequently detected bacterial species (86% and 70% detection, respectively) in bacterial cultures from the subgingiva of cats with periodontal disease [38]. Comparatively, in dogs with chronic ulcerative stomatitis, the microbiome of the ulcerative lesion was also dominated by *Porphyromonas* spp., along with *Neisseria weaveri*, *Fusobacterium* spp., and a bacterium with a *Tannerella forsythia*-like phylotype [17].

More experimental work also points to *P. gulae* as being likely involved in the pathogenesis of FCGS. A study found that out of eight *Porphyromonas* species isolated from companion animals, only *P. gulae* contained virulence and proteolytic activity comparable to that of the human-associated *Porphyromonas gingivalis* [39]. Genes encoding the gingipains proteases RgpA/B and Kgp were identified in the *P. gulae* strain, and induced levels of alveolar bone resorption similar to those caused by *P. gingivalis* [39]. The *P. gulae* strain also exhibited higher capacity for auto-aggregation and binding to oral epithelial cells than all other *Porphyromonas* species examined. These lines of evidence strongly suggest that *P. gulae* has the capacity to contribute to widespread oral inflammation and ulceration in cats with FCGS.

Regarding the other dominant bacteria, both *B. pyogenes* and *F. russii* appear to be residents of the subgingiva or oral flora of cats and dogs [40,41]. A close relative of *F*. *russii*—*F*. *nucleatum*—is an opportunistic oral pathogen that is known to facilitate the development of dental plaque due to its ability to co-aggregate with a diversity of bacteria [42], particularly with *T. forsythia*. In our dataset, FCGS microbiomes were colonized by *T*. *forsythia* but at low abundances (average abundance of 1.36%). Likewise, this microbe is found at a low prevalence in dental swab samples from cats and dogs with periodontal disease [43].

Another well-known periodontal disease-associated pathogen, *Treponema denticola* [44,45], was similarly found at low abundances (1.12%) in our dataset. However, a high relative abundance may not be required for this microbe to negatively impact the oral microbiome. *T. denticola* has been shown to adhere to fibroblasts and epithelial cells, as well as to extracellular matrix proteins present in periodontal tissues [46]. One of its virulence factors (dentilisin) enables invasion and destruction of dental tissues [15,47]. This protease complex is also involved in biofilm formation, degradation of host proteins, and evasion of the host’s immune system [15].

Interestingly, in our study, cats with FCGS did not appear to be heavily colonized by *Pasteurella multocida*, which was previously published as being more prevalent in the caudal oral mucosa of FCGS cats compared with clinically healthy cats [48]. However, this prior study did not report the exact abundance of *P. multocida* and used different methodologies compared with the current study.

### 4.2. Microbiome Diversity in Cats with FCGS Compared with Healthy Cats

We found that the plaque microbiomes of cats with chronic gingivostomatitis were less diverse than those of clinically healthy cats. This is consistent with observations from a study that used 16S rRNA gene sequencing and culture-dependent methods to survey the oral microbiomes of cats with FCGS and healthy cats [48]. Similarly, cats with tooth resorption have less diverse supragingival plaque microbiomes than healthy animals [49]. However, other studies report that cats with FCGS have more diverse microbiomes than healthy cats [11]. In humans with oral lichen planus (OLP)—which may represent a parallel disease process to FCGS—OLP patients had more diverse buccal mucosal microbiomes than healthy controls [50]. Yet other studies conducted in dogs [51], cats [52], and cattle [53] report no significant differences in the gingival microbiome diversity in healthy animals and animals with periodontal disease or gingivitis. Thus, it appears that findings regarding microbiome diversity between healthy and diseased patients are mixed.

### 4.3. Potential Bacterial Drivers of Microbiome Dysbiosis in Cats with FCGS

Our study identified 21 bacterial species (e.g., *Bergeyella zoohelcum*, *Canibacter oris*, *Conchiformibius kuhniae*, *Porphyromonas pasteri*, and *Streptobacillus felis*) and 27 bacterial genera (e.g., *Flavobacterium*, *Moraxella*, *Conchiformibius*, *Neisseria*, *Bergeyella*, *Streptococcus*, *Capnocytophaga*, *Catonella*, *Actinobacillus*, and *Actinomyces*) as being underrepresented in the oral microbiomes of cats with FCGS compared with those of healthy cats. That is, these bacteria were enriched in the oral microbiomes of clinically healthy animals. This indicates that FCGS dysbiosis is characterized by reduced abundances of beneficial or commensal bacteria, rather than the overgrowth of pathogens like *T. forsythia*, *T. denticola*, or *P. gulae*.

A prior study also found a much larger number of bacterial species enriched in healthy cats (39 bacterial species) than in cats with FCGS (16 bacterial species) [12]. Clinically healthy cats had a significantly higher number of aerobic bacteria, such as *Moraxella* spp., *Pseudoclavibacter* spp., *B. zoohelcum*, *Flavobacterium* spp., and *Flavobacteriaceae* spp. Our findings also mirror those from another study that reported that healthy cats had greater abundances of *Moraxella*, *Bergeyella*, *Corynebacterium*, *Capnocytophaga, Actinobacillus,* and *Actinomyces* compared with cats with gingivostomatitis [11,54]. Another study reported that the number of operational taxonomic units (OTUs) for these same bacterial genera (*Moraxella*, *Capnocytophaga*, and *Bergeyella*) were slightly higher in healthy cats compared with cats with tooth resorption [49].

Some of the bacteria strongly associated with the oral microbiomes of healthy cats could be bacteria aiding in digestion. *Capnocytophaga* spp., for example, can ferment carbohydrates to produce short-chain fatty acids (SCFAs) like succinate and acetate [55]. SCFAs are sources of energy for host colonocytes and may play important roles in gene expression and cell signaling [56]. Other bacteria like *P. pasteri* may compete metabolically with pathogenic *Porphyromonas* species like *P. gulae* [57,58]. Another contribution of the commensal bacteria could be as pioneer species. *Neisseria* spp., for example, are early colonizers of plaque biofilms and facilitate the growth of later-comers [59].

Interestingly, the anaerobic and nonmotile *Flexilinea flocculi* was the only bacterial species associated with gingivostomatitis in our dataset. The veterinary dental research community has long wondered whether there were missing spirochetal infections in FCGS. Oral spirochetes in general are thought to possess strong proteolytic activity which allows them to be able to penetrate tissue [60]. The first isolation and characterization of *F*. *flocculi* occurred from sampling methanogenic granular sludge, and revealed that coculturing with hydrogenotrophic methanogens enhanced its growth [61]. In the dental and rumen microbiomes of cattle with periodontitis [62], network analyses have pointed to *Flexilinea* as being highly connected to other bacteria in the network. The authors hypothesized that cattle with periodontitis may produce more methane than healthy animals. A transcriptomic or proteomic analysis of canine and feline *F. flocculi* would yield further insights regarding its potential connection to the development of stomatitis or periodontal disease. Perhaps the combination of *F. flocculi* and *T. denticola* in inflamed oral tissues may play an important role in the progression of FCGS.

It is important to note that our dataset indicated that microbiome dysbiosis was not only present in suspected areas such as gingival and mucosal sites, but also in less studied sites like the tongue and molar salivary gland. Thus, the caudal oral mucosa/palatoglossal arch site does not in and of itself accurately represent the disease entity, at least in terms of the microbiome. Other sites can also be surveyed for the investigation of the etiopathogenesis of oral diseases like FCGS.

### 4.4. Differences between the Microbiomes of Cats with Gingivostomatitis, Tooth Resorption, and/or Periodontitis

We found that the oral microbiome differed between cats with gingivostomatitis, tooth resorption, periodontitis, and a combination of these conditions. This may be expected given that each condition is characterized by its own set of clinical signs, cofactors, and etiological agents. While it is beyond the scope of this publication to discuss the pathogenesis of these oral conditions, they can occur concurrently with FCGS and impact prognosis. In a survey of 75 cats with FCGS [63], 84% of cats had periodontal attachment loss, 24% had furcation involvement, and 14% had mobile teeth. Eighty-nine percent of these cats also had TR, averaging 4.6 lesions per cat. It is thus fascinating that the microbiomes in each separate condition were significantly different.

Another significance of this finding is that it suggests that the etiopathogenesis and molecular mechanisms of stomatitis, periodontitis, or tooth resorption might not be the same, and thus require their own research and treatment. Unfortunately, due to sample size issues, our study was not able to identify the key bacterial species or genera that distinguish the different oral conditions. We strongly encourage future studies to recruit a larger number of cats to address this question.

### 4.5. Swabs of the Oral Cavity Are a Reliable Method of Sampling the Feline Microbiome

We found that microbiomes obtained from swabbing the oral cavity approximated those from plaque. That is, swab samples were able to capture the diversity and variation present in microbiomes that used more direct sampling methods like endodontic paper points. This was concluded after observing a high degree of concordance between plaque and swab microbiomes from the same individual. This type of sampling was also able to capture the microbiome variation among cats with different oral conditions. This is significant because it validates sterile swabs as an appropriate method for sampling the oral cavities of felines.

These results are promising given that swabbing the oral cavity is a go-to method for veterinarians that are sampling the microbiome or testing for oral pathogens. However, we do acknowledge that there is some loss of information and specificity by using swab samples over plaque samples. Nonetheless, sampling using swabs may allow veterinarians to sample a greater number of cats and in a shorter time frame. Plaque can still be used to sample specific regions of the oral cavity like the periodontal pocket.

### 4.6. Limitations

Though we have taken a novel approach to studying the microbiomes of cats with different oral conditions, a few limitations are inherent in this study. First, our sample sizes for cats with tooth resorption and periodontitis were limited to six and five cats, respectively. The number of cats with FCGS and no other dental disease was limited to two cases. Some of the study’s findings might change or be enriched with a larger sample size. Furthermore, because over 60 samples did not contain enough microbial biomass and DNA sequences to be included in the analyses, certain statistical comparisons were not possible. We were unable to identify the bacterial genera that distinguished the oral conditions from each other. We strongly advise future studies to recruit and sample a larger number of cats if they wish to gain a more comprehensive understanding of the microbiome and its role in FCGS.

Related to this, there was a large amount of heterogeneity in the number of sequences yielded by our samples (100 reads–177,000 reads). The technique for sampling using endodontic paper points makes it difficult to control the amount of biofilm collected. Samples collected with a heavier hand may have yielded more sequences.

Unfortunately, we did not discriminate between cats with aggressive periodontitis and chronic periodontitis, which do tend to possess distinct oral microbiomes. Nor did we examine different forms of FCGS, whether erosive or proliferative, and how they impact the microbiome. We encourage future studies to sample a larger number of cats with different oral conditions and different forms of each condition to disentangle how disease dynamics differ between groups.

### 4.7. Future Directions

Recent work depicts the importance of interleukin-6 (IL-6) in the immunoinflammatory pathogenesis of FCGS [10]. Next, we need to understand if associations exist between changes in cytokine levels and oral microbial communities. For example, are the abundances *Bergeyella zoohelcum*, *Capnocytophaga canimorsus*, *Conchiformibius kuhniae*, and *Streptobacillus felis* negatively correlated with particular cytokines? Related to this, studies correlating the oral microbiome with various treatments are needed. Specifically, does the microbiome change with administration of corticosteroids, antimycotics, stem cell therapy, probiotics, or certain antimicrobials? And does this impact clinical outcomes? An investigation of the fecal microbiomes or oral mycobiomes of cats with FCGS, periodontal disease, or no oral disease would be insightful as well.

Finally, this study utilized full-length sequences of the 16S rRNA gene to gain insight into the oral microbiomes of cats with stomatitis, but provided no information about which genes are expressed and their roles in oral disease. The functional characteristics of the microbiome, which can be identified by examining transcripts, proteins, and metabolites, should be considered. Correlations between bacterial composition, metabolic function, immunopathogenic mechanisms, complement system [64], biomarkers, and cytokines will be valuable as well.

An investigation is underway to document the microbiome in feline oral squamous cell carcinoma. A study of the functional characteristics of the microbiota will further enhance the information we have collected. In the future, if we look at both the immune–epithelial compartment and the microbe compartment within the FCGS microenvironment, we may discover tremendous opportunities to target and optimize disease outcomes.

## 5. Conclusions

Our understanding of FCGS as a disease entity has grown over the past few years, but many questions remain unanswered. This evaluation, which establishes the existence of microbiome dysbiosis in different geographical sites, signatures of the microbiome associated with different and concurrent inflammatory disease states, and a role for spirochetes in FCGS, addresses key unknown questions. The novel information we report can contribute to developing improved methods for the prevention, diagnosis, and treatment of this challenging condition in felines.

## Figures and Tables

**Figure 1 animals-13-03544-f001:**
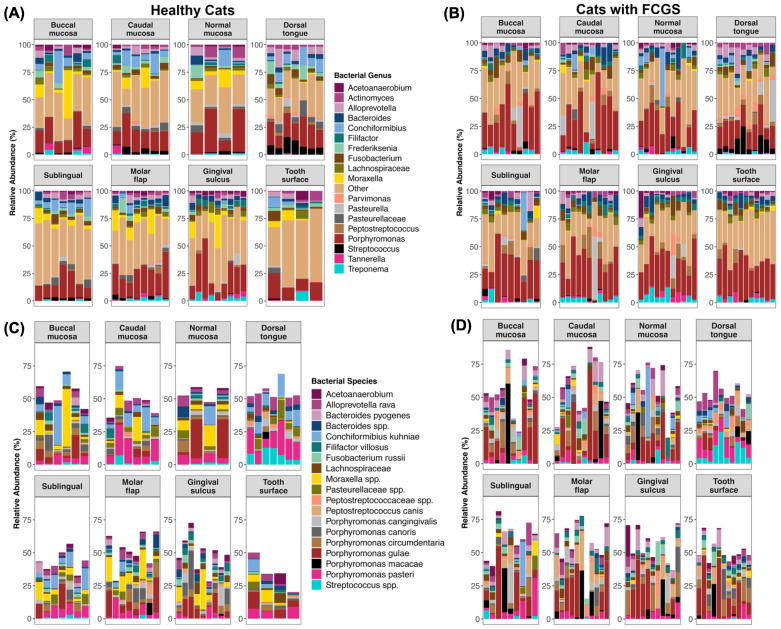
Biogeography of the oral microbiomes in healthy cats (**left**) and cats with FCGS (**right**). Note that microbiome composition from the deep periodontal pocket and external tooth resorption site are excluded. Plots showing the relative abundances of bacterial genera are on top (**A**,**B**) and of bacterial species at the bottom (**C**,**D**). Bacterial genera with mean relative abundances <1.28% are encapsulated by the “Other” category.

**Figure 2 animals-13-03544-f002:**
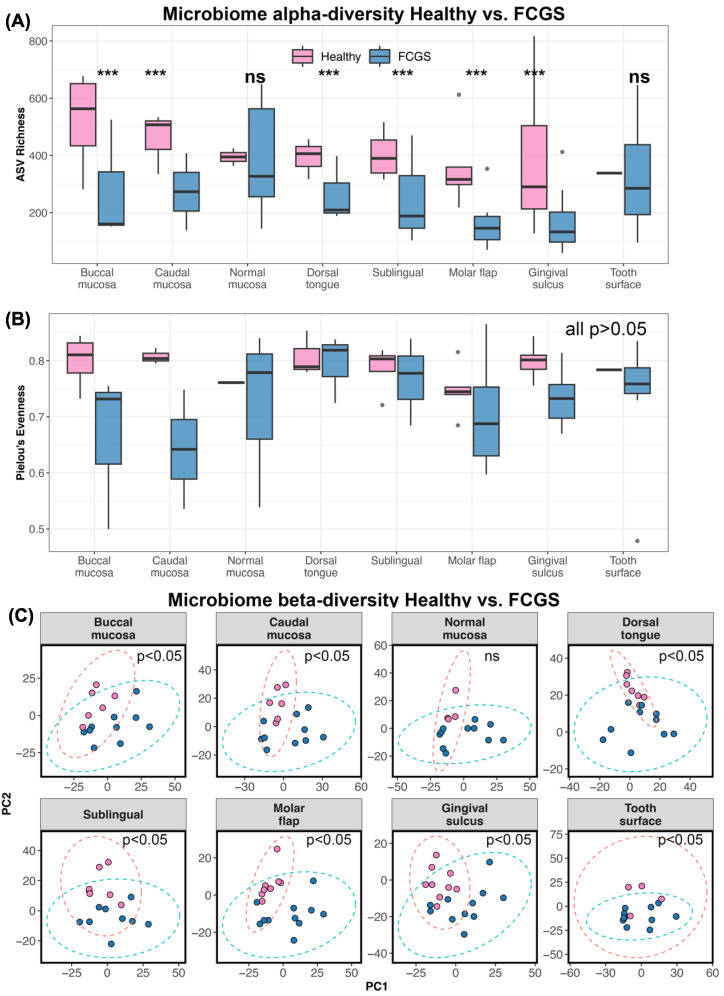
The oral microbiomes of cats with FCGS are distinct from those of healthy cats. (**A**) Boxplots of microbiome ASV richness and (**B**) Pielou’s evenness in plaque samples from eight oral sites, in healthy cats and cats with FCGS. Asterisks *** indicate statistical significance (*p* = 0.05), ns denotes where difference was not statistically significant. (**C**) PCA ordinations based on Aitchison distances calculated from bacterial species abundances. PERMANOVA *p*-values are shown.

**Figure 3 animals-13-03544-f003:**
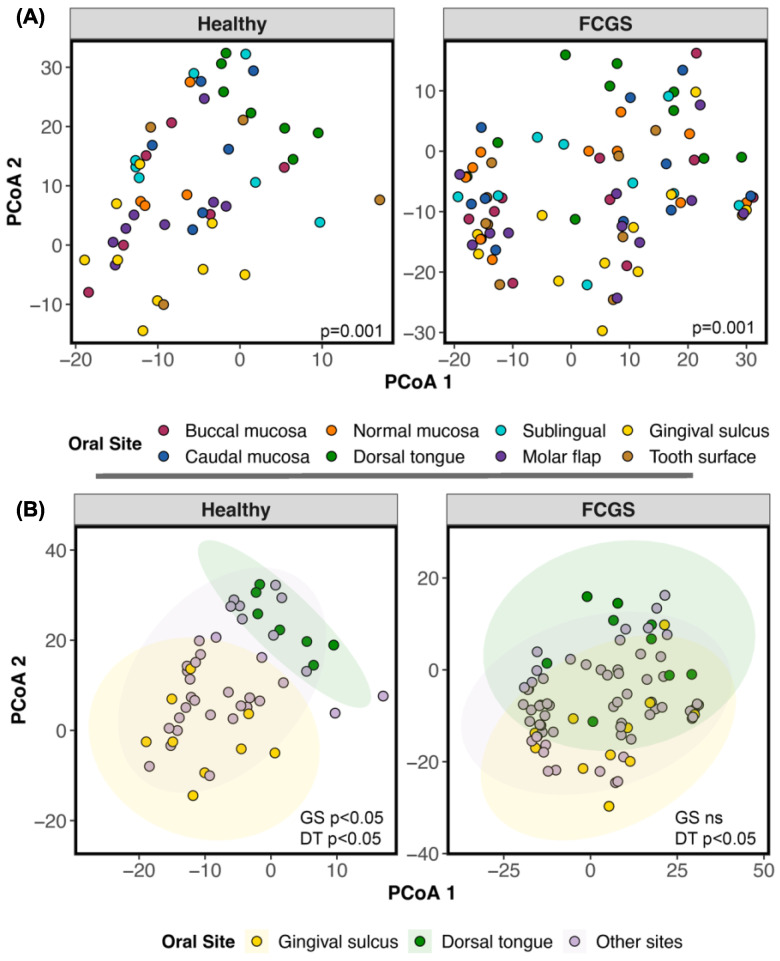
The microbiomes of the dorsal tongue and gingival sulcus cluster separately from the microbiomes at other oral sites. (**A**) PCA ordinations based on Aitchison distances calculated from species-level bacterial abundances, color-coded by oral site. Healthy cats are on the left and cats diagnosed with FCGS are on the right. (**B**) The same ordination but meant to highlight the separation of two sites (dorsal tongue; gingival sulcus) from the rest.

**Figure 4 animals-13-03544-f004:**
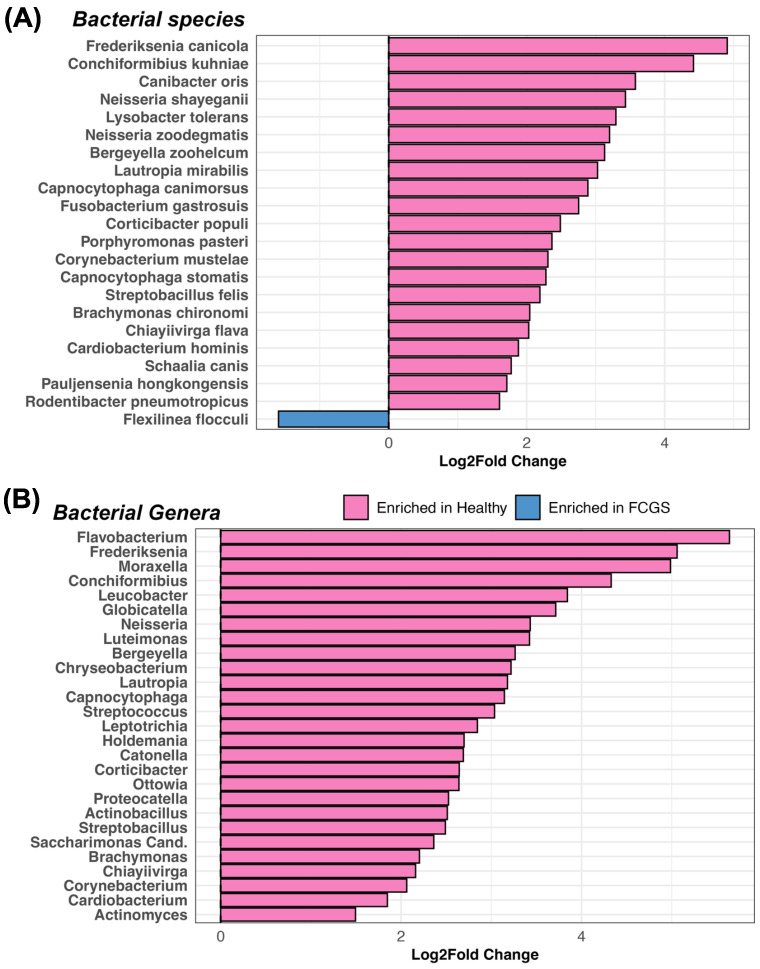
The oral microbiomes of cats with FCGS have an underrepresentation of 21 bacterial species compared with the oral microbiomes of healthy cats. Results from LinDA differential abundance analyses performed at the (**A**) bacterial species or (**B**) bacterial genus level. Bacterial taxa enriched in healthy cats are in pink while those enriched in FCGS cats are in blue.

**Figure 5 animals-13-03544-f005:**
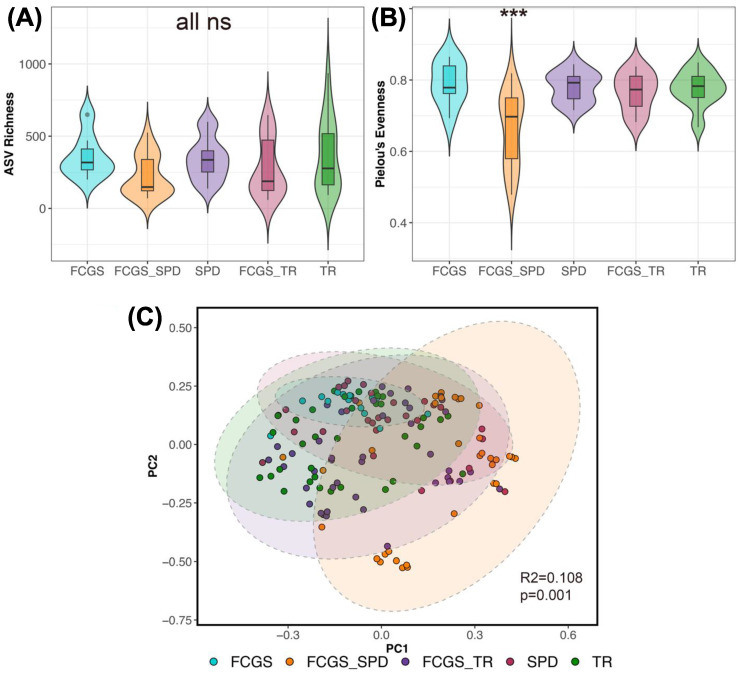
Microbiome varies with oral condition in felines. (**A**) Microbiome ASV richness and (**B**) Pielou’s evenness for plaque samples from cats diagnosed with stomatitis (FCGS), tooth resorption (TR), periodontitis (SPD), or a combination of these conditions. Samples were pooled across oral sites for analysis with GLMMs (Appendix A). Asterisks *** indicate statistical significance (*p* = 0.05), ns denotes where difference was not statistically significant. (**C**) PCoA ordination based on Bray–Curtis distances showing the clustering of oral microbiome samples by condition. PERMANOVA output is shown.

**Figure 6 animals-13-03544-f006:**
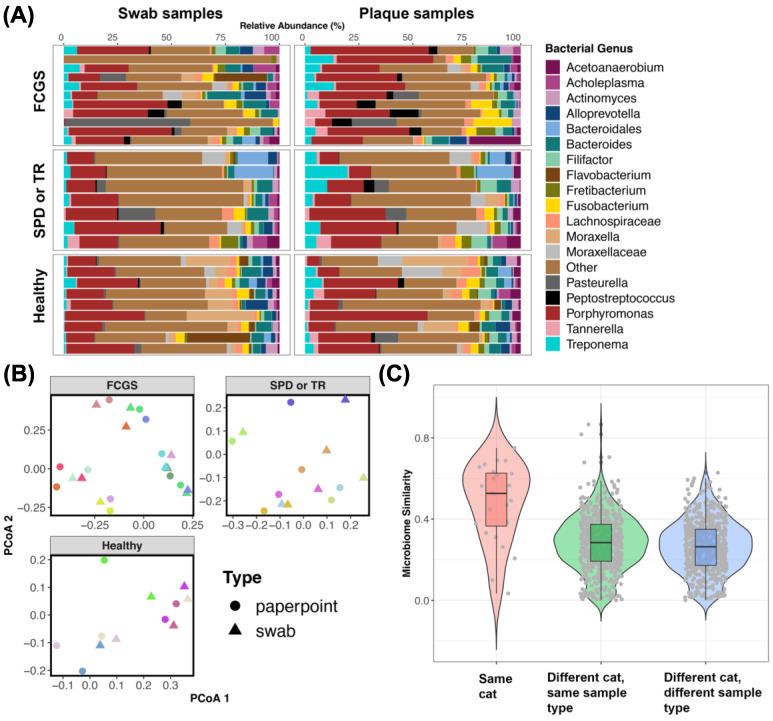
Oral swab microbiomes are comparable to plaque microbiomes. (**A**) Relative abundances of bacterial genera in swab samples taken from the oral cavity and in plaque samples taken from the gingival sulcus using endodontic paper points. Bacteria genera with mean relative abundances < 1.27% are represented by the “Other” category. (**B**) PCoA ordinations based on Bray–Curtis distances. Samples are color-coded by host identity to showcase the similarity between the two sample types. (**C**) Microbiome similarity (1-Bray–Curtis dissimilarity) between pairs of samples from the same cat or different cats.

**Table 1 animals-13-03544-t001:** Median age, sex, and body weight of healthy cats and cats affected with severe periodontitis (SPD), tooth resorption (TR), or feline chronic gingivostomatitis (FCGS).

	Healthy (*n* = 9 but 3 NAs)	SPD (*n* = 5)	TR (*n* = 6, but 1 NA)	FCGS (*n* = 12, but 1 NA)	All (*n* = 32 but 5 NAs)
Age (years)	4.6	5	9	5	6.4
Sex (% male)	43	60	40	45	46
Body weight (kg)	4.4	5.5	5.5	4.6	4.75

NA = not available; three of the nine healthy cats had missing (e.g., NA) values for age, sex, body weight, and spay or neuter status. One of the cats with TR had missing metadata and one of the cats with FCGS also had missing metadata.

**Table 2 animals-13-03544-t002:** Microbiomes of swab samples from the oral cavity vary with host sex in felines.

	Swabs of Oral Cavity		Plaque Samples of the Gingival Sulcus	
Predictor	Bray–Curtis (*n* = 25)	Aitchison (*n* = 25)	Bray–Curtis (*n* = 25)	Aitchison (*n* = 25)
Age (years)	R^2^ = 0.042, *p* = 0.31	R^2^ = 0.03, *p* = 0.83	R^2^ = 0.04, *p* = 0.29	R^2^ = 0.036, *p* = 0.63
Sex (M/F)	R^2^ = 0.15, *p* = 0.01	R^2^ = 0.136, *p* = 0.012	R^2^ = 0.035, *p* = 0.67	R^2^ = 0.041, *p* = 0.46
Body weight (kg)	R^2^ = 0.049, *p* = 0.15	R^2^ = 0.04, *p* = 0.407	R^2^ = 0.43, *p* = 0.39	R^2^ = 0.042, *p* = 0.4

Marginal PERMANOVA models correlating microbiome beta diversity with host age (years), sex (M/F; all were spayed or neutered), and body weight (kg). Bacterial species-level abundances were used for calculation of distance matrices. Cats with missing metadata or swab samples were excluded. Statistically significant results are highlighted in bold.

## Data Availability

Raw PacBio HiFi sequences have been deposited to the NCBI Sequence Read Archive (SRA) (accessions SAMN37669375-SAMN37669612), under BioProject PRJNA1023696. Sample metadata, the ASV count table, and tables from statistical tests are in the Appendix A.

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
