# Peer review of "The Oral Microbiome across Oral Sites in Cats with Chronic Gingivostomatitis, Periodontal Disease, and Tooth Resorption Compared with Healthy Cats"

_animals, 2023, doi:10.3390/ani13223544_

Round 1

Reviewer 1 Report

Comments and Suggestions for Authors

This manuscript by Anderson el al, aims to describe the oral microbiome assessed in different sites and in different conditions compared to healthy (non oral condition) cats. This is an interesting study contributing towards better understanding of a common frustrating disease amongst domestic cats.

In general I found the paper very interesting and very thorough in therms of all the data you produced. A weakness or a limitation is the number of cases with individual diseases like cats with FCGS were only 2 of them. The results might differ if you tested more cats. The different sample collection methods and areas is very interesting and eye opening too. However the variation of microbiome is likely to change in other area down the digestive tract too, not just the oral cavity. I understand however your study is describing the oral microbiome. Overall is a great paper that identify and covers a gap or at least warrants further studies to cover that gap. 

More specific comments 

L 58. You probably want to describe location of "lesions", as stomatitis in itself would comprehend inflammation +/- ulceration, proliferation..etc

L98 and 110 table 1. That information should not be in the introduction but in the material and methods section if so detailed. 

L155. That paragraph and in other parts of the section M&M (i.e 157, 159, 163...). When describing a product there are inconsistencies. Product, city, country, if USA product, city, state. Sometimes missing city and state, sometimes using USA but not city and state...etc. Review how to report this info with journal.

L 498. This is applicable to other parts of the manuscript. Sometimes just the genus is reported, sometimes one species of the genus is implied (sp.) sometimes various species of the genus are used (spp.) just wondering if this needs to be reviewed. First time the genus is reported expand it, then contract next  (when specific species are mentioned) unless is the start of a sentence. Like in L79, 494, 497, 508, 510, 516 for instance Porphyromonas gulae are expanded every time. After L79 the rest could be contracted.

L557 -560. All species of bacteria would need to be itallics. 

L 564. As above

L 580. Spell out at the beggining of sentence. 

L. 607. In that sentence Peralta is reporting something not comparable. Is reporting caudal oral mucosa is more susceptible to chronic inflammation (we see that as clinically evident in the majority of FCGS). You state that in terms of microbiome does not represent the disease entity (I agree just in terms of microbiome). These are very different statements in my opinion. Microbiome population versus clinical disease....we observe histopathology in other areas where you interestingly found dysbiosis but the tissue is normal. 

Comments on the Quality of English Language

Although I am not qualified for commenting on this I think is well written and there are am not really significant issues. I still detected some minor typos and edits that would need to be addressed I encourage authors to go through the manuscript and review thoroughly before publishing

Reviewer 2 Report

Comments and Suggestions for Authors

Chronic Gingivostomatitis, Severe Periodontal Dis-ease, and Tooth Resorption Compared to a Healthy Population " by Jamie G. Anderson et al., Ref: animals-2679484,

provides data feline chronic Gingivostomatitis (FCGS). The pathogenesis of FCGS is yet unclear. FCGS is a chronic mucosal and gingival inflammatory disease affecting cats. This pathogenesis involves interactions between the host's inflammatory process, its immune response, and the oral microbiome. The authors collected sterile non-invasive plaque biofilm samples from ten distinct sites in the oral cavity 37 of cats with stomatitis (n=12), healthy cats (n=9), and cats with tooth resorption or periodontitis 38 (n=11) to gain a better understanding of this disease and the influence of the microbiome on host inflammatory disease states. they discovered that the microbiomes of cats with FCGS exhibited significant dysbiosis at 39 oral sites. Additionally, beta-diversity of the microbiome varied with oral condition, indicating that stoma-40 titis, periodontitis, and/or tooth resorption influence the microbiome differently. Finally, they discovered that the microbiomes of oral cavity swabs were comparable to those extracted from plaque using endodontic paper points, validating it as an alternative sampling technique. Collectively, the advances the understanding of the dysbiosis and bacterial composition of the oral microbiome in feline gingivostomatitis (FCGS), with the goal of contributing to the prevention, diagnosis, and treatment of this challenging condition.

Overall, the technical and scientific sound of the manuscript seems reasonable.

After careful evaluation, I have some minor concerns regarding the publication of this article in the journal Animals.

Herein you can find the comments regarding the manuscript:

1) The title seems a bit too long

2) The English language should be deeply revised. Typos, syntax, and vocabulary should be carefully checked throughout the text.

3) The section "Materials and methods" is very detailed.

4) The discussion of the data and conclusion should be better structured. Some parts are very long-winded and difficult to follow.

Comments on the Quality of English Language

The English language should be deeply revised. Typos, syntax, and vocabulary should be carefully checked throughout the text.

Reviewer 3 Report

Comments and Suggestions for Authors

The manuscript studies the impact of oral microbiome in the pathogenesis of the Feline Chronic Gingivostomatitis (FCGS), an inflammatory disease that affects oral mucosa and gingiva in cats.  Oral microbiome at several biogeographical sites was studied, looking for eventual differences among them.  Oral microbiome was compared between healthy cats and individual suffering from FCGS. Also, oral microbiome was studied in cat suffering from other oral diseases, Severe periodontal Disease (SPD) and Tooth Resorption (TR), and compared to that of cats with FSGC. Finally, two types of sampling methods were compared (endodontic paper points and swabs of oral cavity). 

  This study is aimed to know if microbiomes from different oral diseases are the same across biogeographical sites, what until now was unknown. The obtained results suggest that the etiopathogenesis of these oral diseases might not be the same.    On the other hand, the comparison of two sampling methods is another novelty of this work; swabs allow for quick and easy sampling of large numbers of cats. This new knowledge could be useful for developing better prevention, diagnosis and treatment for these diseases and therefore, this article would be very interesting for specialists in feline dentistry.

 However, as it is affirmed in Limitations, small sample sizes prevented from identifying the key bacterial species or genera that distinguish the different oral conditions. Also, no information is provided about which genes   from the 16S rRNA region are expressed and their roles in oral disease, as said in Limitations.

 The molecular and statistical methodologies are appropriate to the objectives and described in detail. The authors discuss adequately the obtained results.

 The figures, tables and supplementary material are very informative and useful for understanding this work.

 The references are recent: 29/65 (44.61%) date from the last 5 years and some are even from this year.

 Specific  comments:

 Please, make  sure that the first time you write an abreviation, its meaning is explained.

 Table 1: there is no indication about the meaning of NA

 Table S1:

 - It is said in the caption:  Metadata associated with all of the microbiome samples utilized in this study. However it is said at the bottom:  Below is information on each sample that was excluded in this study. Is”excluded” a correct word? Would you mean “included”?

-Sorry , I cannot find the column Total_Num samples

 Two references seem to be incomplete:

  38 . Gaskell, R.M.; Tj, G.J. Intractable Feline Stomatitis. Veterinary Annual.

64.  Anderson, J.G. FCGS Case Series with Radiographic Review. In Proceedings of the Proceedings; 2006.
